# A contractile injection system stimulates tubeworm metamorphosis by translocating a proteinaceous effector

Charles F Ericson[1,2,3†], Fabian Eisenstein[3†], João M Medeiros[3], Kyle E Malter[1,2], Giselle S Cavalcanti[1,2], Robert W Zeller[1], Dianne K Newman[4,5], Martin Pilhofer[3*], Nicholas J Shikuma[1,2*]

[1]Department of Biology, San Diego State University, San Diego, United States; [2]Viral Information Institute, San Diego State University, San Diego, United States; [3]Department of Biology, Institute of Molecular Biology and Biophysics, Eidgenössische Technische Hochschule, Zürich, Switzerland; [4]Division of Biology and Biological Engineering, California Institute of Technology, Pasadena, United States; [5]Division of Geological and Planetary Sciences, California Institute of Technology, Pasadena, United States

**Abstract** The swimming larvae of many marine animals identify a location on the sea floor to undergo metamorphosis based on the presence of specific bacteria. Although this microbe–animal interaction is critical for the life cycles of diverse marine animals, what types of biochemical cues from bacteria that induce metamorphosis has been a mystery. Metamorphosis of larvae of the tubeworm *Hydroides elegans* is induced by arrays of phage tail-like contractile injection systems, which are released by the bacterium *Pseudoalteromonas luteoviolacea*. Here we identify the novel effector protein Mif1. By cryo-electron tomography imaging and functional assays, we observe Mif1 as cargo inside the tube lumen of the contractile injection system and show that the *mif1* gene is required for inducing metamorphosis. Purified Mif1 is sufficient for triggering metamorphosis when electroporated into tubeworm larvae. Our results indicate that the delivery of protein effectors by contractile injection systems may orchestrate microbe–animal interactions in diverse contexts.
DOI: https://doi.org/10.7554/eLife.46845.001

*For correspondence:
pilhofer@biol.ethz.ch (MP);
nshikuma@sdsu.edu (NJS)

†These authors contributed equally to this work

## Introduction

Bacteria can have profound effects on the normal development of diverse animal taxa (*McFall-Ngai et al., 2013*). One of the most pervasive examples of bacteria stimulating development is the induction of animal metamorphosis by bacteria (*Hadfield, 2011*). During these interactions in marine environments, surface-bound bacteria often serve as environmental triggers that induce mobile animal larvae to settle on a surface and undergo metamorphosis. Although the stimulation of metamorphosis by bacteria is critical for diverse animal-mediated processes such as coral reef formation (*Webster et al., 2004*; *Whalan and Webster, 2014*), the recruitment of stocks for marine fisheries (*Dworjanyn and Pirozzi, 2008*; *Yu et al., 2010*) and the fouling of submerged surfaces like the hulls of ships (i.e. biofouling) (*Khandeparker et al., 2006*; *Nedved and Hadfield, 2008*), we know little about the mechanisms that govern this microbe–animal interaction.

Despite the fact that the link between bacteria and animal metamorphosis was first discovered in the 1930s (*Zobell and Allen, 1935*), few bacterial products have been described that stimulate this developmental transition. To date, identified bacterial cues can all be classified as small molecules. Two examples are the small bacterial metabolite tetrabromopyrrole, which induces partial or

**eLife digest** Many marine animals, including corals and tubeworms, begin life as larvae swimming in open water before transforming into adults that anchor themselves to the seabed. These transformations, known as metamorphoses, are often triggered by certain types of bacteria that form friendly relationships (or "symbioses") with the animals.

One such symbiosis forms between a bacterium called *Pseudoalteromonas luteoviolacea* and a tubeworm known as *Hydroides elegans.* Previous studies have shown that *P. luteoviolacea* produces syringe-like structures known as Metamorphosis Associated Contractile structures (or MACs for short) that are responsible for stimulating metamorphosis in the tubeworm larvae. Some viruses that infect bacteria use similar structures to inject molecules into their host cells. However, it was not clear whether MACs were also able to inject molecules into cells.

Here, Ericson, Eisenstein et al. used a technique called cryo-electron tomography combined with genetic and biochemical approaches to study how the MACs of *P. luteoviolacea* trigger metamorphosis in tubeworms. The experiments identified a protein in the bacteria named Mif1 that was required for the tubeworms to transform. The bacteria loaded Mif1 into the tube of the MAC structure and then injected it into the tubeworms. Further experiments showed that inserting Mif1 alone into tubeworms was sufficient to activate metamorphosis.

Mif1 is the first protein from bacteria to be shown to activate metamorphosis, but it is likely that many more remain to be discovered. Since other marine animals also form symbioses with bacteria, understanding how Mif1 and other similar proteins work may inform efforts to restore coral reefs and other fragile ecosystems, and increase the production of oysters and other shellfish. Furthermore, MACs and related structures may have the potential to be developed into biotechnology tools that deliver drugs and other molecules directly into animal cells.
DOI: https://doi.org/10.7554/eLife.46845.002

complete metamorphosis of corals (*Sneed et al., 2014*; *Tebben et al., 2011*) and the polar molecule histamine from algae or associated microbes, which induces urchin metamorphosis (*Swanson et al., 2007*). To our knowledge, however, no proteinaceous bacterial cues have yet been identified that stimulate animal metamorphosis.

To investigate how bacteria induce animal metamorphosis, we have previously studied the interaction between the tubeworm *Hydroides elegans* (hereafter *Hydroides*) and the bacterium *Pseudoalteromonas luteoviolacea* (*Hadfield et al., 1994*; *Huang and Hadfield, 2003*; *Nedved and Hadfield, 2008*; *Shikuma et al., 2016*). We found that *P. luteoviolacea* produces arrays of Metamorphosis Associated Contractile structures (MACs) that induce the metamorphosis of *Hydroides* larvae (*Huang et al., 2012*; *Shikuma et al., 2014*). MACs are an example of a Contractile Injection System (CIS); macromolecular machines that are specialized to puncture membranes and often deliver proteinaceous effectors into target cells (*Brackmann et al., 2017*; *Taylor et al., 2018*). Like other CISs, MACs are evolutionarily related to the contractile tails of bacteriophages (bacterial viruses) and are composed of an inner tube protein (homologous to gp19 from phage T4 and Hcp from type six secretion systems) surrounded by a contractile sheath, a tail-spike, and a baseplate complex (*Shikuma et al., 2014*). The conserved mechanism is driven by contraction of the sheath, which propels the inner tube/spike into the target cell.

Different ways of loading effectors onto a CIS have been suggested. Translocation mechanisms of effectors via the spike complex of a CIS have been well characterized (*Quentin et al., 2018*; *Shneider et al., 2013*). The presence of an alternative pathway of loading effectors into the inner tube lumen has been speculated but is only poorly understood. For example, an amorphous density inside the inner tube of the Antifeeding prophage (Afp) was attributed to either the toxin payload or a tape-measuring protein (*Heymann et al., 2013*). Other classes of effectors were found to interact with the inner tube protein (Hcp) and are likely released post-firing by tube dissociation in the target cytoplasm (*Sana et al., 2016*; *Silverman et al., 2013*).

In this study, we set out to identify a potential metamorphosis-inducing effector that MACs inject into *Hydroides* larvae, as well as the loading of such an effector into MACs. We show that a previously identified genomic region in *P. luteoviolacea* (*Shikuma et al., 2016*) encodes a bacterial

protein that localizes to the inner tube lumen of the MAC structure and is necessary for inducing the metamorphosis of *Hydroides* larvae. Our results identify a proteinaceous effector stimulating animal metamorphosis and provide a direct visualization of an effector in the tube lumen of an assembled CIS.

## Results

### Two bacterial genes are responsible for densities within the inner tube lumen of MACs and are involved in metamorphosis induction

We previously identified a genomic locus in *P. luteoviolacea* encoding six genes (gene numbers JF50_12590, JF50_12595, JF50_12600, JF50_12605, JF50_12610 and JF50_12615) that was essential for inducing the larvae of *Hydroides* to undergo metamorphosis (*Shikuma et al., 2016*). Here we analyzed biofilms of strains with in-frame deletions of each of the six genes and tested their ability to induce *Hydroides* metamorphosis. The ΔJF50_12605 and ΔJF50_12615 mutants exhibited a reduced ability to induce metamorphosis (less than 20%, *Figure 1A*), while mutation of the other four genes had no observable effect. When JF50_12605 and JF50_12615 were replaced back into their native chromosomal loci, metamorphosis induction was restored (*Figure 1B*). We confirmed the effect of MACs on *Hydroides* metamorphosis by producing cell-free MAC array preparations. While larvae exposed to MACs from a ΔJF50_12615 mutant did not induce metamorphosis (even at high concentrations), MACs from a ΔJF50_12605 mutant induced metamorphosis when added at higher doses (*Figure 1—figure supplement 1A/B*). Our results suggest that JF50_12615 was essential for the induction of metamorphosis, while JF50_12605 contributed but was dispensable. Based on our results here and below, we name the protein encoded by JF50_12615 as 'Mif1' for Metamorphosis-Inducing Factor 1.

To search for structural differences between MACs from wildtype *P. luteoviolacea* and the specific gene deletion mutants, we employed cryo-electron tomography (cryoET) imaging. Deletion of the full JF50_12590–JF50_12615 locus (*Figure 1—figure supplement 2*), or each of the six genes individually (*Figure 1—figure supplement 3*), did not impair the formation of ordered arrays of MACs, featuring both extended and contracted conformations. Upon detailed analyses, we observed that extended MACs from both ΔJF50_12605 and Δ*mif1* strains exhibited a central lumen with very low density. By contrast, MACs from wildtype and the other deletion mutants possessed a density distribution that was homogeneous and a lumen was not discernable (*Figure 1C–G* and *Figure 1—figure supplement 3*). We refer to these structural phenotypes as 'empty' and 'filled' respectively. Strikingly, quantitative analyses showed that the empty phenotype in ΔJF50_12605 and Δ*mif1* MACs correlated with the inability to induce metamorphosis (*Figure 1A/H*). The replacement of *mif1* and JF50_12605 back into their native chromosomal loci reverted the empty phenotype back to filled (*Figure 1—figure supplement 4*).

### The density within the MAC tube lumen represents a cargo protein

To investigate whether the structural differences between wildtype and Δ*mif1*/ΔJF50_12605 MACs represented potential cargo, we performed sub-tomogram averaging of the extended sheath-tube complex (resolution estimation in *Figure 2—figure supplement 1*). The resulting MAC structures for both wildtype and Δ*mif1* revealed densities corresponding to the sheath and the inner tube (*Figure 2A–F*), similar to the structures of homologous CISs (*Jiang et al., 2019*). While the Δ*mif1* structure lacked any discernible density inside the ~4 nm-wide tube lumen (*Figure 2E,F*), the wildtype structure exhibited repeating packets of density inside the tube (*Figure 2A–D*), suggesting the presence of a potential cargo. The densities in the tube lumen reinforced less strongly compared to the sheath-tube complex, which could be caused by one or a combination of the following factors: 1) during averaging, the alignment of the cargo was affected by the strong densities from the sheath-tube, 2) the cargo was not structured or flexible, and/or 3) the cargo was present at a substoichiometric amount compared to tube subunits or tube-rings. In any case, it is likely that the packet-like shape of the cargo density was caused by alignment artifacts. This is supported by a difference map between the wildtype and Δ*mif1* structure, which shows a continuous density filling the inner tube lumen (*Figure 2G,H*). It is important to note, however, that the cargo and tube densities were separated by a low-density region (arrowheads in *Figure 2B*). Furthermore, any expelled tubes

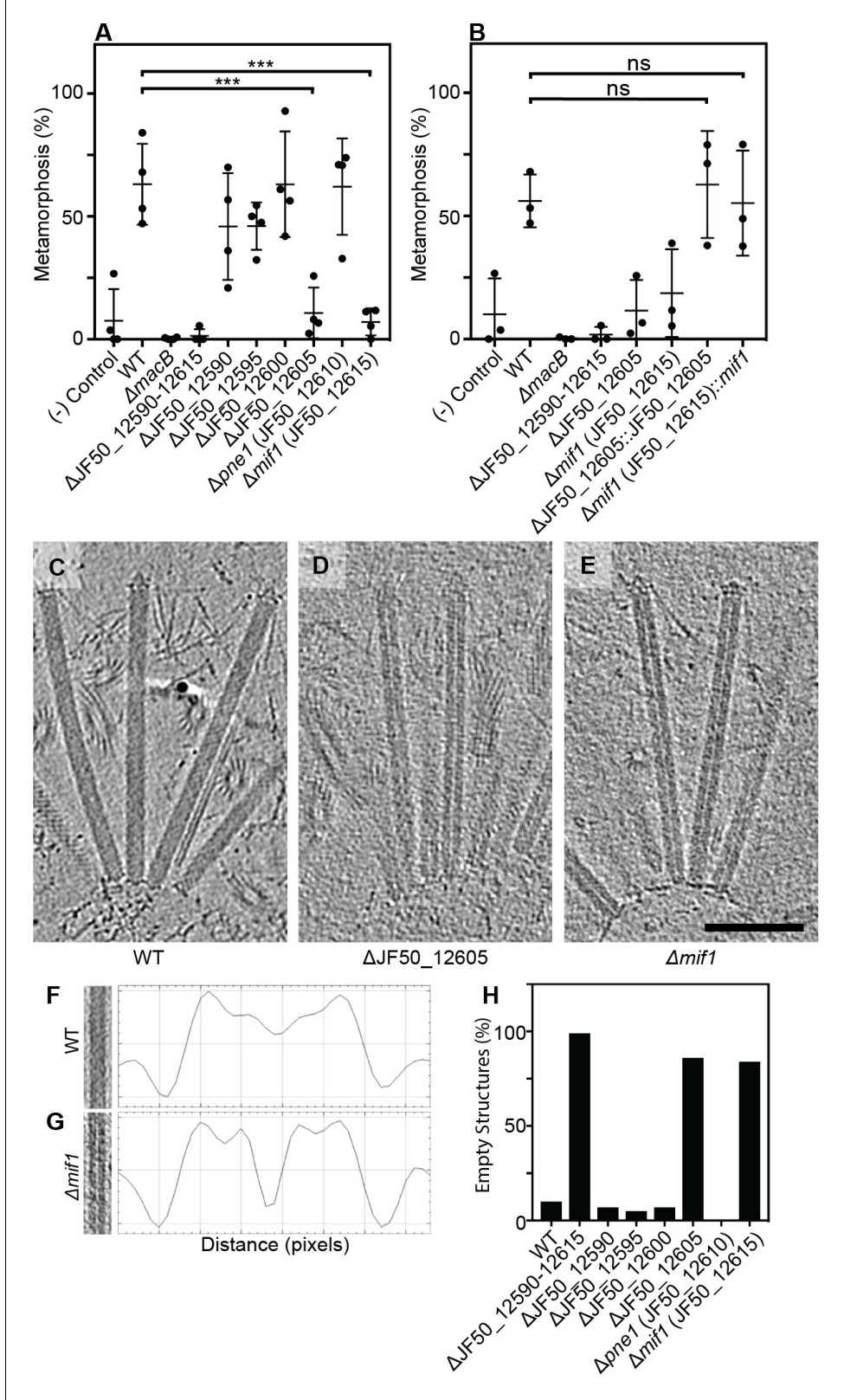

**Figure 1.** Two bacterial genes are important for inducing *Hydroides* metamorphosis and deletions of these genes produce MACs with an 'empty' phenotype. (**A**) Metamorphosis (%) assay of *Hydroides* larvae in response to biofilms of *P. luteoviolacea* wildtype (WT) and different gene deletion strains. Deletion of JF50_12605 or *mif1*

*Figure 1 continued on next page*

*Figure 1 continued*

(JF50_12615) showed a significant loss in the ability to induce metamorphosis when compared to wildtype. (**B**) Restoration of JF50_12605 and *mif1* (JF50_12615) into their native chromosomal loci restored function. Graphs in (**A/B**) show an average of biological replicates, where each point represents one biological replicate. *p-value≤0.05, ns = not significant. (**C–E**) Representative cryotomographic images of the 'filled' phenotype from wildtype MACs (**C**), and 'empty' phenotype from ΔJF50_12605 (**D**), and Δ*mif1* (**E**) MACs. Scale bar, 100 nm. (**F/G**) Shown are representative MAC structures (on left; taken from **C/E**) and their density plots. The wildtype 'filled' phenotype shows a relatively homogeneous density profile across the diameter of the MAC. The Δ*mif1* 'empty' phenotype shows a low-density region in the center of the MAC. (**H**) Shown is the fraction of empty structures for different deletion mutants as observed by cryoET imaging. Note that the 'empty' phenotype correlates with the inability to induce metamorphosis (**A**).

DOI: https://doi.org/10.7554/eLife.46845.003

The following figure supplements are available for figure 1:

**Figure supplement 1.** Metamorphic response of *Hydroides* larvae to cell-free MAC extracts from wild type *P. luteoviolacea* and individual gene mutants.
DOI: https://doi.org/10.7554/eLife.46845.004
**Figure supplement 2.** Wildtype and ΔJF50_12590-12615 have structurally similar arrays.
DOI: https://doi.org/10.7554/eLife.46845.005
**Figure supplement 3.** 'Filled' and 'empty' phenotypes in all studied gene mutant strains.
DOI: https://doi.org/10.7554/eLife.46845.006
**Figure supplement 4.** Replacing JF50_12605 and *mif1* into their native chromosomal loci generates MACs with filled tubes.
DOI: https://doi.org/10.7554/eLife.46845.007

from triggered MACs always showed an 'empty' phenotype in cryotomography images (*Figure 2—figure supplement 2*). These results together could indicate weak or entirely absent interactions between cargo and tube, possibly facilitating rapid release of the cargo from the tube upon contraction.

To test whether JF50_12605 and/or Mif1 were present within the MAC complex and represented the cargo within the tube lumen, we performed protein identification by mass spectrometry of purified MACs. In two independent experiments, we detected Mif1 but not JF50_12605 in wildtype MAC samples (*Figure 3A*). MACs from the ΔJF50_12605 mutant exhibited considerably fewer spectral counts for the Mif1 protein. These results are consistent with the 'empty' phenotype observed by cryoET imaging of the ΔJF50_12605 strain (*Figure 1D* and *Figure 1—figure supplement 3E*). To further corroborate the association of Mif1 with MAC arrays, we tagged Mif1 with a FLAG-tag in its native chromosomal locus in three different locations. After purifying MACs, we detected Mif1 strongly associated with MACs from two Mif1-FLAG-tagged strains and one at a reduced level by dot-blot and an anti-FLAG antibody (*Figure 3B*).

Because JF50_12605 is important for localizing Mif1 within the MAC complex, we analyzed protein–protein interactions between JF50_12605 and Mif1. To this end, we performed a reciprocal pull down of S-tagged JF50_12605 and 6xHis-tagged Mif1. We detected JF50_12605 when pulling down Mif1 by nickel chromatography, and we detected Mif1 when pulling down JF50_12605 with S-tag antibodies (*Figure 3C*). To determine whether Mif1 or JF50_12605 associated with other components of the MAC complex, we utilized a bacterial two-hybrid system based on the interaction-mediated reconstruction of a cyclic AMP (cAMP) signaling cascade (*Karimova et al., 2000*). When JF50_12605, Mif1 and MacT1 (JF50_12680, tube) were screened for interactions, we found a significant interaction between JF50_12605 and Mif1 as well as JF50_12605 with itself (*Figure 3D–F*). However, neither JF50_12605, nor Mif1 interacted with MacT1 (JF50_12680, tube). Together, these data indicate that Mif1 is present within the MAC structure and represents the densities seen in the tube lumen, while JF50_12605 might act as a chaperone that helps to localize Mif1 inside the MAC tube. Mif1, however, could also associate with MACs independently of JF50_12605 in an inefficient manner. This is shown by 1) the residual presence of Mif1 in MACs from a ΔJF50_12605 mutant as detected by mass spectrometry (*Figure 3A*), and 2) the fact that high concentrations of a cell-free ΔJF50_12605 MAC extract can induce metamorphosis (*Figure 1—figure supplement 1*).

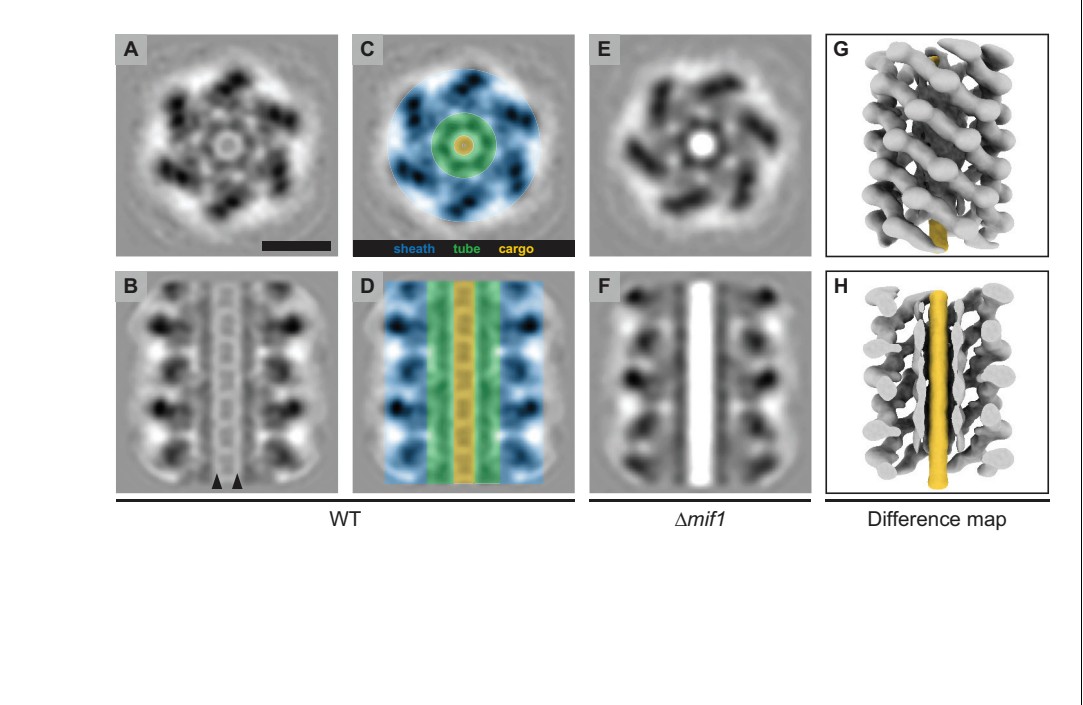

**Figure 2.** MACs from a Δ*mif1* mutant lack electron density in the tube lumen. (A–F) Cross sectional (A/C/E) and longitudinal (B/D/F) slices through subtomogram averages of the MAC sheath-tube complex from wildtype (WT; A–D) and Δ*mif1* (E–F). The hexameric sheath and tube modules could be clearly discerned (indicated in C/D). The inner tube lumen displayed clear differences in density between WT and Δ*mif1*. The wildtype tube lumen was filled with densities that likely represent cargo (A-D, indicated in yellow), which was not present in the Δ*mif1* lumen (E/F). Note the low-density region that separates the tube and cargo (indicated by arrowheads in B). (G/H) Shown are isosurfaces of the Δ*mif1* structure (gray) and of a difference map (yellow; calculated from the wildtype and Δmif1 structure), highlighting the additional density in the wildtype tube lumen. Scale bar, 10 nm.
DOI: https://doi.org/10.7554/eLife.46845.008

The following figure supplements are available for figure 2:

**Figure supplement 1.** Fourier shell correlations.
DOI: https://doi.org/10.7554/eLife.46845.009

**Figure supplement 2.** Triggered MAC tubes show 'empty' phenotype.
DOI: https://doi.org/10.7554/eLife.46845.010

## Purified and electroporated Mif1 protein induces tubeworm metamorphosis

Because our results suggested that Mif1 was loaded into the MAC tube lumen, we next tested whether Mif1 was sufficient for stimulating metamorphosis when delivered to *Hydroides* larvae. We therefore purified N-terminally His-tagged Mif1 by nickel chromatography (*Figure 4A*) and verified its identity by western blot with a Mif1-specific antibody (*Figure 4B*). As controls, we purified JF50_12605 and GFP under the same conditions. Mif1 protein provided exogenously to competent larvae of *Hydroides* at concentrations of up to 250 ng/μl did not stimulate metamorphosis (*Figure 4—figure supplement 1*). We reasoned, however, that the Mif1 protein might require intracellular delivery into host cells to initiate metamorphosis of *Hydroides*. To this end, we utilized a custom electroporator that was previously successfully used for other marine invertebrates (*Zeller, 2018*; *Zeller et al., 2006*). Successful translocation of protein was confirmed by anti-GFP western blotting of larval lysate after electroporation (*Figure 4—figure supplement 2*). When we delivered Mif1 into competent *Hydroides* larvae, a significant percentage of the larvae underwent metamorphosis (*Figure 4C*). In contrast, neither JF50_12605 nor GFP stimulated metamorphosis when electroporated under the same conditions. Our results suggest that Mif1 was sufficient to stimulate *Hydroides* metamorphosis when delivered by electroporation.

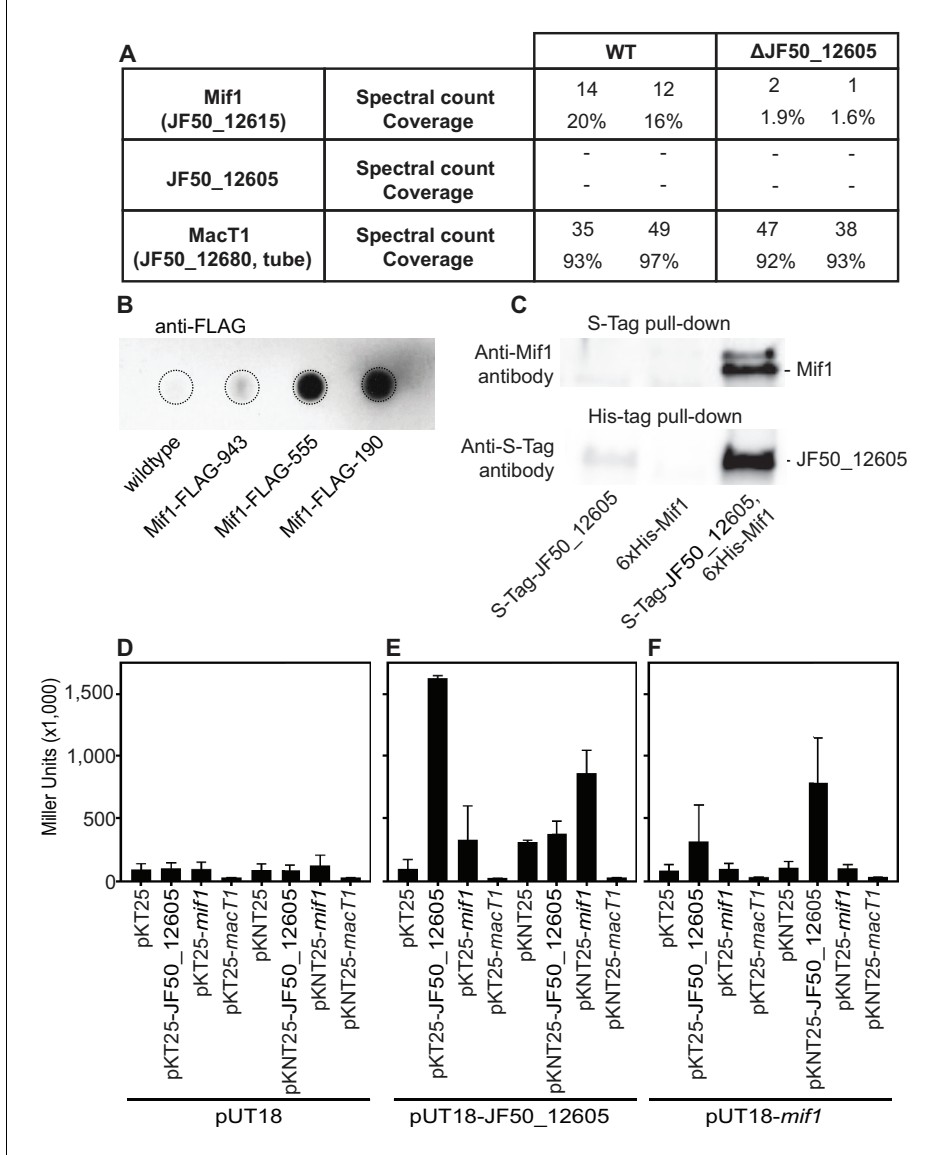

**Figure 3.** Mif1 is present in MAC complexes and JF50_12605 is required for Mif1's association with the MAC complex. (**A**) Mass spectrometry of wildtype MAC arrays detected Mif1 but not JF50_12605. Spectral counts for Mif1 were low for MACs purified from the ΔJF50_12605 mutant, indicating a possible chaperone-like function for JF50_12605. (**B**) Dot blot of purified MACs from wildtype or strains with Mif1 tagged at amino acid positions 943 [C-terminus] (Mif1-FLAG-943), 555 (Mif1-FLAG-555), and 190 (Mif1-FLAG-190) were probed with anti-FLAG antibody. The signal indicates association of Mif1 with MAC arrays. (**C**) Co-expression, reciprocal pull-down and western blotting of S-tagged JF50_12605 and 6xHis-tagged Mif1 indicate an interaction between both proteins. Strains in which only one component was tagged were used as controls. (**D–F**) Quantification of bacterial two-hybrid experiments were used to analyze possible interactions between JF50_12605, Mif1 and tube (MacT1) proteins. Briefly, the two fragments of CyaA (T18/T25) were fused to the respective target proteins with the CyaA activity only being restored by interaction between target proteins. JF50_12605 showed a strong interaction with itself and with Mif1.
DOI: https://doi.org/10.7554/eLife.46845.011

## Discussion

In conclusion, our data indicate that the bacterium *P. luteoviolacea* induces metamorphosis of animal larvae by delivering the effector protein Mif1 via the tube lumen of an extracellular contractile injection system (MACs). These insights are significant for different fields of research as discussed below.

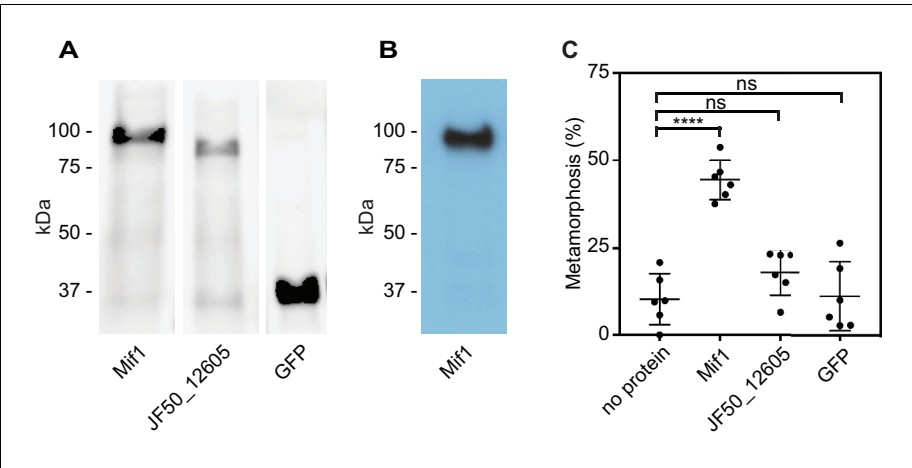

**Figure 4.** Mif1 is sufficient for stimulating metamorphosis when delivered by electroporation. (**A**) Shown is an SDS page gel of purified Mif1, JF50_12605 and GFP. (**B**) Western blot of purified Mif1 protein probed with a C-terminal anti-Mif1 peptide antibody confirms Mif1 identity. (**C**) Metamorphosis (%) of *Hydroides* larvae 24 hr after electroporation with purified Mif1, JF50_12605 or GFP protein, shows induction of metamorphosis by electroporated Mif1. Graph shows an average of biological replicates, where each point represents one biological replicate. ****p-value≤0.0001 by t-test, ns = not significant.

DOI: https://doi.org/10.7554/eLife.46845.012

The following figure supplements are available for figure 4:

**Figure supplement 1.** Purified Mif1 is unable to induce metamorphosis when added exogenously.
DOI: https://doi.org/10.7554/eLife.46845.013

**Figure supplement 2.** Quantification of GFP protein associated with larvae after electroporation.
DOI: https://doi.org/10.7554/eLife.46845.014

First, previously reported bacterial cues for animal metamorphosis are all classified as small molecules (*Sneed et al., 2014*; *Swanson et al., 2007*; *Tebben et al., 2011*). The identification of a protein (Mif1) that stimulates tubeworm metamorphosis requires us to expand the scope of possible biomolecules and mechanisms by which bacteria stimulate animal development. Our findings suggest that rather than MACs stimulating metamorphosis solely by physical puncturing and depolarization of larval membranes, the delivered Mif1 protein could have an enzymatic activity. This challenges previous hypotheses on metamorphosis induction (*Carpizo-Ituarte and Hadfield, 1998*; *Yool et al., 1986*), however, it would be in line with many studies on other eukaryote-targeting CIS effectors with enzymatic activities (*Jiang et al., 2016*; *Ma and Mekalanos, 2010*; *Vlisidou et al., 2019*). Future investigations will shed light on the molecular mechanism by which Mif1 triggers animal metamorphosis, which is a challenging task, given that Mif1 does not feature any recognizable conserved protein domains.

Second, our results directly showed the previously hypothesized possibility of effector delivery via the tube lumen of a CIS (*Heymann et al., 2013*; *Sana et al., 2016*; *Shneider et al., 2013*; *Silverman et al., 2013*). Interestingly, the comparison of MACs with a different class of CIS, namely the Type Six Secretion System (T6SS), reveals significant differences. The T6SS effectors that are thought to be delivered by the T6SS tube lumen show protein–protein interactions between the T6SS effector and the T6SS tube protein (Hcp) (*Sana et al., 2016*; *Silverman et al., 2013*). By contrast, we did not detect such interactions between Mif1 and MAC tube protein. One possible explanation could be that the biophysical characteristics of the T6SS tube and the MAC tube are different. While the T6SS tube is inherently unstable and disassembles soon after contraction (*Szwedziak and Pilhofer, 2019*), inner tubes of MACs and other extracellular CISs (and contractile phages) can be readily detected by electron microscopy and therefore seem to be much more stable. Given our observation that expelled MAC tubes were always empty (e.g. *Figure 2—figure supplement 2*), this poses the question of how the effectors exit such a stable tube after contraction. We hypothesize that this could be the very reason for weak or entirely absent interactions between Mif1 and MAC tube, as well as for the low-density region that was seen in subtomogram averages

separating Mif1 and MAC tube (*Figure 2B*). Another mechanistic consequence of low affinity between Mif1 and tube could be the requirement of an assembly factor, that is JF50_12605, that allows for efficient targeting of Mif1 to the tube.

Third, our insights into MAC function could be significant to re-engineer the system for future medical and biotechnological applications. As micron-scale, syringe-like structures, MACs have potential for being developed as delivery systems targeting eukaryotic cells. Extracellular CISs such as MACs are of particular interest, because they are released from the producing bacterial cell and autonomously bind to the target cell's surface. Extracellular CISs that target bacterial pathogens are already under development as narrow host-range antimicrobial agents (*Scholl, 2017*). The identification of effectors carried by MACs provides the basis for loading MACs with a cargo of choice. We recently reported a second effector that MACs deliver to eukaryotic cells (*Rocchi et al., 2019*), expanding the number of MAC effectors that could be engineered. Intriguingly, a cryotomogram of the Δ*pne1* (JF50_12610) mutant shows that the tube has a filled phenotype (*Figure 1—figure supplement 3*). The filled phenotype suggests that Pne1 is not found within the inner tube, but instead could be loaded in a different location (e.g. spike) within the MACs complex. Understanding tube lumen-delivered effectors could be particularly helpful, based on the potential of a higher payload per CIS, as compared to spike-bound effectors.

Fourth, the identification of Mif1 and its delivery mechanism will facilitate the investigation of how bacterial factors trigger animal signaling systems, leading to metamorphosis. This could have potential practical applications for preventing biofouling, improving aquaculture husbandry, restoring degraded ecosystems like coral reefs, and as a biotechnology platform.

The fact that bacteria are known to stimulate metamorphosis in every major group of animals alive today (*Hadfield, 2011*), combined with the detection of MAC-like gene clusters in microbes from diverse environments including the ocean, terrestrial environments and even the human gut (*Böck et al., 2017*; *Jiang et al., 2019*), underscores the huge diversity of bacterium–animal interactions that remains to be explored.

## Materials and methods

### Metamorphosis assays

Bioassays were conducted with specimens of *Hydroides elegans* obtained from Quivira Basin, San Diego, California. Embryos were obtained and maintained as previously described (*Nedved and Hadfield, 2008*; *Shikuma et al., 2016*). Competent larvae were exposed to biofilms of *P. luteoviolacea* wild type, as a positive control, to *P. luteoviolacea* mutants, and to *P. luteoviolacea* strains unable to produce MAC structures (Δ*macB*), as well as to artificial seawater (-). The percent of larvae that underwent metamorphosis was scored 24 hr after the induction of metamorphosis. Metamorphosis was scored visually by observing the number of individuals that formed branchial radioles, and a primary and secondary tube. Four biological replicates of approximately 30 larvae each were performed for each treatment on three separate occasions with larvae spawned from different adults.

### Bacterial strains, plasmid construction and culture conditions

All bacterial strains, plasmids and primer sequences used are listed in the supplemental *Tables 1* and *2*. All deletion and fusion strains were created according to previously published protocols (*Rocchi et al., 2019*; *Shikuma et al., 2016*; *Shikuma et al., 2014*). Plasmid insert sequences were verified by DNA sequencing. Deletion and insert strains were confirmed by PCR. All *E. coli* strains were grown in Lysogeny-Broth (LB) media at 37°C shaking at 200 revolutions per minute (RPM). All *P. luteoviolacea* cultures were grown in seawater tryptone (SWT) media (35.9 g/l Instant Ocean, 2.5 g/l tryptone, 1.5 g/l yeast extract, 1.5 ml/l glycerol) at 25°C shaking at 200 RPM. Media that contained antibiotics were at a concentration of 100 mg/ml unless otherwise stated.

### Gentle MAC extraction

*P. luteoviolacea* was grown in 50 ml SWT media in 250 ml flasks at 30 °C for 6 hr or overnight (12–14 h). Cells were centrifuged for 30 min at 4000 g and 4 °C and resuspended in 5 ml cold extraction buffer (20 mM Tris, pH 7.5, 1M NaCl). Cultures were centrifuged for 30 min at 4000 g and 4 °C and

**Table 1.** Strains and plasmids used in this work.

| Strain no. | | Genotype | Source or reference |
|---|---|---|---|
| | **Strain** | | |
| NJS5 | HI1 Str$^R$ | *P. luteoviolacea* HI1, Str$^R$ | (*Huang et al., 2012*) |
| NJS23 | Δ*macB* | *P. luteoviolacea* HI1, Str$^R$ Δ*macB* | (*Shikuma et al., 2014*) |
| NJS235 | ΔJF50_12590-F50_12615 | *P. luteoviolacea* HI1, Str$^R$ ΔR4 | (*Shikuma et al., 2016*) |
| NJS289 | ΔJF50_12590 | *P. luteoviolacea* HI1, Str$^R$ ΔJF50_12590 | This Study |
| NJS287 | ΔJF50_12595 | *P. luteoviolacea* HI1, Str$^R$ ΔJF50_12595 | This Study |
| NJS285 | ΔJF50_12600 | *P. luteoviolacea* HI1, Str$^R$ ΔJF50_12600 | This Study |
| NJS283 | ΔJF50_12605 | *P. luteoviolacea* HI1, Str$^R$ ΔJF50_12605 | This Study |
| NJS281 | ΔJF50_12610 | *P. luteoviolacea* HI1, Str$^R$ ΔJF50_12610 | This Study |
| NJS279 | ΔJF50_12615 | *P. luteoviolacea* HI1, Str$^R$ ΔJF50_12615 | This Study |
| NJS294 | ΔJF50_12605::12605 | *P. luteoviolacea* HI1, Str$^R$ ΔJF50_12605::JF50_12605 | This Study |
| NJS295 | ΔJF50_12615::12615 | *P. luteoviolacea* HI1, Str$^R$ ΔJF50_12615::JF50_12615 | This Study |
| | **Plasmid** | | |
| pNJS007 | pCVD443 | Amp$^R$, Km$^R$, sacB, pGP704 derivative | (*Huang et al., 2012*) |
| pNJS266 | pCVD443_Δ12590 | pCVD443::Δ12590 Amp$^R$, Km$^R$ | This Study |
| pNJS265 | pCVD443_Δ12595 | pCVD443::Δ12595 Amp$^R$, Km$^R$ | This Study |
| pNJS264 | pCVD443_Δ12600 | pCVD443::Δ12600 Amp$^R$, Km$^R$ | This Study |
| pNJS263 | pCVD443_Δ12605 | pCVD443::Δ12605 Amp$^R$, Km$^R$ | This Study |
| pNJS262 | pCVD443_Δ12610 | pCVD443::Δ12610 Amp$^R$, Km$^R$ | This Study |
| pNJS261 | pCVD443_Δ12615 | pCVD443::Δ12615 Amp$^R$, Km$^R$ | This Study |
| pNJS256 | pCVD443_12590–615 complement | Amp$^R$, Km$^R$ | (*Shikuma et al., 2016*) |
| pNJS282 | pCVD443_12605 complement | Amp$^R$, Km$^R$ | This Study |
| pNJS074 | pCVD443_Δ12585 | Amp$^R$, Km$^R$ | This Study |
| pNJS267 | pUT18 | Amp$^R$ | (*Karimova et al., 2000*) |
| pNJS268 | pUT18C | Amp$^R$ | (*Karimova et al., 2000*) |
| pNJS269 | pKT25 | Km$^R$ | (*Karimova et al., 2000*) |
| pNJS270 | pKNT25 | Km$^R$ | (*Karimova et al., 2000*) |
| pNJS283 | pUT18_12605 | Amp$^R$ | This Study |
| pNJS299 | pUT18_12615 | Amp$^R$ | This Study |
| pNJS527 | pUT18_12680 | Amp$^R$ | This Study |
| pNJS284 | pKT25_12605 | Km$^R$ | This Study |
| pNJS285 | pKT25_12615 | Km$^R$ | This Study |
| pNJS529 | pKT25_12680 | Km$^R$ | This Study |
| pNJS286 | pKNT25_12605 | Km$^R$ | This Study |
| pNJS300 | pKNT25_12615 | Km$^R$ | This Study |
| pNJS530 | pKNT25_12680 | Km$^R$ | This Study |
| pNJS393 | pET15b_12605 | Amp$^R$ | This Study |

*Table 1 continued on next page*

*Table 1 continued*

| Strain no. | | Genotype | Source or reference |
| --- | --- | --- | --- |
| pNJS395 | pET15b_12615 | Amp[R] | This Study |
| pNJS397 | pET15b_GFP | Amp[R] | This Study |

DOI: https://doi.org/10.7554/eLife.46845.015

the supernatant was isolated and centrifuged for 30 min at 7000 g and 4 ˚C. The pellet was resuspended in 20–100 µl cold extraction buffer and stored at 4˚C for further use.

## Plunge freezing of MACs

Plunge freezing was performed as implemented in *Weiss et al. (2017)*. In essence, gentle MAC extractions were seeded with 10 nm BSA-coated colloidal gold particles (1:4 v/v, Sigma) and 4 µl of the mixture was applied to a glow-discharged holey-carbon copper EM grid (R2/1, Quantifoil). The grid was backside blotted in a Vitrobot (FEI Company) by using a Teflon sheet on the front pad, and plunge-frozen in a liquid ethane-propane mixture (37%/63%) cooled by a liquid nitrogen bath. Frozen grids were stored in liquid nitrogen.

## Cryo-electron tomography

The gentle MAC extractions were imaged by cryo-electron tomography (cryoET) (*Weiss et al., 2017*). Images were recorded on a Titan Krios TEM (FEI) equipped with a Quantum LS imaging filter operated at a 20 eV slit width and K2 Summit (Gatan). Pixel sizes at specimen level ranged from 2.14 Å to 2.72 Å. Tilt series were collected using a bidirectional tilt-scheme from −30˚ to +60˚ and −32˚ to −60˚ in 2˚ increments. Total dose was ~90 e-/Å$^2$ and defocus was kept at −5 to −6 µm. Some tilt series were recorded in focus using a Volta phase plate (*Danev et al., 2014*). Tilt series were acquired using SerialEM (*Mastronarde, 2005*) and reconstructed and segmented using the IMOD program suite (*Kremer et al., 1996*). Density plots to determine filled and empty phenotypes were done using Fiji (*Schindelin et al., 2012*). Contrast enhancement of some tomograms was done using the tom_deconv deconvolution filter (https://github.com/dtegunov/tom_deconv).

## Sub-tomogram averaging

Tomograms used for structure identification and picking were binned by a factor of 4. Defocus was estimated using Gctf (*Zhang, 2016*) and CTF correction, exposure filtering and backprojection was done using IMOD. SR data were binned by a factor of 2 resulting in a pixel size of 4.29 Å/px. The discrete extended MAC structures were identified visually in individual tomograms and their longitudinal axes were modeled with open contours in 3dmod (*Mastronarde, 2008*). Individual model points were added at defined intervals of about 12 nm along the contours using the addModPts program from the PEET package (*Heumann et al., 2011*) resulting in 24'721 initial particles for filled tubes and 37'024 initial particles for empty tubes. Models were imported into Dynamo (*Castaño-Díez et al., 2012*), particles were extracted, the azimuth angle was randomized and all particles were averaged to obtain an initial reference. Four times binned subtomograms (17.14 Å/px) were used for four iterations of initial alignment with the reference low-pass filtered to about 50 Å. No symmetry was applied, but rotational search was limited to +/- 30˚. Before unbinning, subtomograms were cleaned by distance and cross correlation coefficient leaving 20'358 filled and 20'782 empty tube particles. Two times binned subtomograms (8.57 Å/px) were extracted using the refined coordinates and the dataset was split in two half sets. Half sets were aligned independently for five iterations. Unbinned subvolumes (4.29 Å/px) were extracted using the refined coordinates and aligned for eight more iterations. Subvolumes were cleaned by cross correlation coefficient and final averages were generated using 13'039 particles and 13'425 particles for filled and empty tubes, respectively. UCSF Chimera (*Pettersen et al., 2004*) was used for visualization of the 3D models and to generate the difference map between wildtype and Δ*mif1* structure.

**Table 2.** Primers used in this work.

| Primer | Sequence |
| --- | --- |
| 1556_dA | TGATGGGTTAAAAAGGATCGATCCTCTAGATTGGAGCAATAAACGGGTTC |
| 1556_dB | GTTCATAATTAAACTGCGATCGCAGCCATAAGGCCTCCTTGATA |
| 1556_dC | TATCAAGGAGGCCTTATGGCTGCGATCGCAGTTTAATTATGAAC |
| 1556_dD | TTTTGAGACACAACGTGAATTCAAAGGGAGAGCTCCGCTTTGGGTACTGGCTTTA |
| 1556_intF | CCGAGCAAACGTTATCACAA |
| 1556_intR | TCAGCGCTCTCATTATGTGC |
| 1555_dA | TGATGGGTTAAAAAGGATCGATCCTCTAGACCGAGCAAACGTTATCACAA |
| 1555_dB | CCTTGCATGAGGTTAAGAAAGTTTGACGTACCCTTCAGCCATATT |
| 1555_dC | AATATGGCTGAAGGGTACGTCAAACTTTCTTAACCTCATGCAAGG |
| 1555_dD | TTTTGAGACACAACGTGAATTCAAAGGGAGAGCTCGATGCGGTAACGGTTGTTCT |
| 1555_intF | AGCGATTGATGCTGAACAAA |
| 1555_intR | ACCATCGCATAACCCGTAAC |
| 1554_dA | TGATGGGTTAAAAAGGATCGATCCTCTAGATACGCCGTCCAGTTAGGACT |
| 1554_dB | GTTTGTTAACGTCACGGCAGCTGCATTGCCATTTAAACTCC |
| 1554_dC | GGAGTTTAAATGGCAATGCAGCTGCCGTGACGTTAACAAAC |
| 1554_dD | TTTTGAGACACAACGTGAATTCAAAGGGAGAGCTCATTGATTGGAAGCGCGATAG |
| 1554_intF | TTTATGAGGCACCAACGACA |
| 1554_intR | GCCTGTGCCGTTTTATCTGT |
| 1553_dA | TGATGGGTTAAAAAGGATCGATCCTCTAGAGGCGATCAGTGGAGTGAAGT |
| 1553_dB | AATACTTCTTGCTCAGCCCCGCGTGCTTCTTCTGTCATGT |
| 1553_dC | ACATGACAGAAGAAGCACGCGGGGCTGAGCAAGAAGTATT |
| 1553_dD | TTTTGAGACACAACGTGAATTCAAAGGGAGAGCTCTCAGAACCAGCAGTCTCACG |
| 1553_intF | CGGGCCTAGAAATCACTCAA |
| 1553_intR | TCGACGTCAAATCAGTCGAG |
| 1552_dA | TGATGGGTTAAAAAGGATCGATCCTCTAGAGAGAGCAAGAAGTGGCGAGT |
| 1552_dB | TAGCCTTTTAGTGCCGCTTTTGAGGCGTCCATATCTGACA |
| 1552_dC | TGTCAGATATGGACGCCTCAAAAGCGGCACTAAAAGGCTA |
| 1552_dD | TTTTGAGACACAACGTGAATTCAAAGGGAGAGCTCTGCTGACCAAGCAGATTGAC |
| 1552_intF | GGGCAATTGTTGTGGATTTT |
| 1552_intR | TGATCCCAAACCACTTGTGA |
| 1551_dA | TGATGGGTTAAAAAGGATCGATCCTCTAGAGACTGCTGGTTCTGATTCGAT |
| 1551_dB | AACAGATCATTACATTAAAATGAGCCTCTGTTCTTGTTGTTGCATTTCA |
| 1551_dC | TGAAATGCAACAACAAGAACAGAGGCTCATTTTAATGTAATGATCTGTT |
| 1551_dD | TTTTGAGACACAACGTGAATTCAAAGGGAGAGCTCCTTCTCCATTTTCGCCTTTG |
| 1551_intF | CGTTTTCAGTGACCATCACG |
| 1551_intR | CGGTGGGCAAAAAGGTATAA |
| pUT18_605_F1 | CAGCTATGACCATGATTACGCCAAGCTTGCATGCCATGACAGAAGAAGCACGCGAAAAAA |
| pUT18_605_R1 | CTGGCGGCTGAATTCGAGCTCGGTACCCGGGGATCATTCACAAGTGCTAATTGATAAAAT |
| pUT18_615_F1 | CAGCTATGACCATGATTACGCCAAGCTTGCATGCCATGCAACAACAAGAACAGGAGCAAG |
| pUT18_615_R1 | CTGGCGGCTGAATTCGAGCTCGGTACCCGGGGATCCATTAAAATGAGCCTTTCTTTTTCA |
| pUT18_680_F | CATGATTACGCCAAGCTTGCATGCCATGGCTACTACTAAAGCAGATATCG |
| pUT18_680_R | AATTCGAGCTCGGTACCCGGGGATCATGGAACTCAATCTTGATGTCATCT |
| pKT_605_F1 | CCGATTACCTGGCGCGCACGCGGCGGGCTGCAGGGATGACAGAAGAAGCACGCGAAAAAA |
| pKT_605_R1 | AACGACGGCCGAATTCTTAGTTACTTAGGTACCCGCTAATTCACAAGTGCTAATTGATAA |

*Table 2 continued on next page*

*Table 2 continued*

| Primer | Sequence |
| --- | --- |
| pKT_615_F1 | CCGATTACCTGGCGCGCACGCGGCGGGCTGCAGGGATGCAACAACAAGAACAGGAGCAAG |
| pKT_615_R1 | AACGACGGCCGAATTCTTAGTTACTTAGGTACCCGTTACATTAAAATGAGCCTTTCTTTT |
| pKT_680_F1 | CCTGGCGCGCACGCGGCGGGCTGCAATGGCTACTACTAAAGCAGATATCG |
| pKT_680_R1 | GCCGAATTCTTAGTTACTTAGGTACTTAATGGAACTCAATCTTGATGTCA |
| pKNT_605_F1 | CAGCTATGACCATGATTACGCCAAGCTTGCATGCCATGACAGAAGAAGCACGCGAAAAAA |
| pKNT_605_R1 | TGATGCGATTGCTGCATGGTCATTGAATTCGAGCTATTCACAAGTGCTAATTGATAAAAT |
| pKNT_615_F1 | CAGCTATGACCATGATTACGCCAAGCTTGCATGCCATGCAACAACAAGAACAGGAGCAAG |
| pKNT_615_R1 | TGATGCGATTGCTGCATGGTCATTGAATTCGAGCTCATTAAAATGAGCCTTTCTTTTTCA |
| pKNT_615_F2 | CATGATTACGCCAAGCTTGCATGCCATGCAACAACAAGAACAGGAGCAAG |
| pKNT25-680_F | CATGATTACGCCAAGCTTGCATGCCATGGCTACTACTAAAGCAGATATCG |
| pKNT25-680_R | GCTGCATGGTCATTGAATTCGAGCTATGGAACTCAATCTTGATGTCATCT |
| pET15b_605_F1 | TGCCGCGCGGCAGCCATATGATGACAGAAGAAGCACGCG |
| pET15b_605_R1 | GCTTTGTTAGCAGCCGGATCCCTAATTCACAAGTGCTAATT |

DOI: https://doi.org/10.7554/eLife.46845.016

## Bacterial two-hybrid analysis

Bacterial two-hybrid Analysis was performed following the protocols detailed previously (*Karimova et al., 2000*). Briefly, proteins of interest were cloned into one of four Bacterial Two Hybrid (BTH) plasmids pUT18, pUT18C, pKT25, and pKNT25. These produced individual N- or C-terminal fusions between the proteins of interest and the T18 and T25 subunits on of the adenylate cyclase (*CyaA*) protein. All plasmid sequences were confirmed by PCR. Plasmid combinations containing the genes of interest were then electroporated into BTH101 electrocompetent cells that lacked a native *CyaA* gene. The BTH101 cells were grown on LB agar containing ampicillin (100 mg/ml), kanamycin (100 mg/ml) and 1% glucose. Glucose was used to suppress the expression of proteins before performing the assay. Protein–protein interactions were quantified by performing a β-galactosidase assay with cells being grown overnight at 37°C and shaking at 200 RPM. Protein expression was induced with 1.0 mM IPTG. The cultures were incubated at 25°C shaking at 200 RPM for 6 hr before being mixed with a one-step 'β-gal' mix (*Schaefer et al., 2016*). A plate reader was then used to measure the absorbance at 420 nm and 600 nm. The optical densities were used to calculate Miller Units as previously described (*Miller, 1972*).

## Protein purification and electroporation

To purify JF50_12615, JF50_12605 and GFP proteins, genes of interest were cloned into the pET15b plasmid and grown in *E. coli* BL21 pLysE. Bacteria were struck out on LB agar plates with ampicillin (100 μg/μl) and grown at 37°C for 24 hr. A single colony was inoculated into 5 ml LB with ampicillin (100 μg/μl) and grown at 37°C shaking at 200 RPM for 14–16 hr. The overnight culture was diluted 1:500 into 500 ml LB with ampicillin (100 μg/μl), grown at 37°C shaking at 200 RPM until the culture reached an $OD_{600}$ of 0.95. Protein expression was induced with 0.1 mM IPTG and grown for 25°C for 16 hr. The culture was centrifuged at 4000 g for 20 min and the supernatant was removed. The pellet was then resuspended in lysis buffer (20 mM imidazole, 25 mM tris-HCl, 500 mM NaCl, pH 8) with a protease inhibitor cocktail (100 μM leupeptin, 1 μM pepstatin and 5 μM bestatin). The culture was French pressed twice (1000 psi) and sonicated 3 times for 10–30 s each time. The lysed culture was then spun down at 12,000 g for 20 min and the supernatant was discarded. Inclusion bodies were purified from the pellet by first washing the pellet twice with 20 mM tris pH 8, 2 M urea, 2% triton X-100, 500 mM NaCl. The remaining pellet was resuspended using 5 ml 6M guanidinium HCl, 5 mM imidazole, 20 mM tris pH 8, 500 mM NaCl. The 6XHIS tagged proteins were then bound to Ni-agarose beads which had been pre-equilibrated to the resuspension buffer. The proteins were refolded by adding 1 ml/min of 5 mM imidazole, 20 mM tris pH 8, 500 mM NaCl up to a total of 50 ml. After refolding, the beads were loaded onto a vacuum column and washed twice with

10 ml of refolding buffer. The protein was then eluted using 250 mM imidazole, 20 mM tris pH 8, 500 mM NaCl. Fractions containing the protein were buffer exchanged into a storage buffer (25 mM tris, 250 mM NaCl, pH 7.6) and stored at −80℃. A Bradford protein assay (BioRad) was done in order to quantify the amount of protein present. An antibody produced against the Mif1-specific peptide sequence CERSKGEFTEGKPKP (Genscript) was used to confirm expression and purification.

## Western blot

Protein samples and lysates were first normalized using Bradford protein assay to quantify protein concentrations. Equal concentrations of protein were loaded onto BioRad stain-Free SDS-PAGE gels 4–20% (catalog no. 4568093) and imaged prior to transfer to confirm equal loading using BioRad gel doc ez system and stain free tray. The protein gel was then used to transfer protein to a PVDF membrane via semi-dry transfer system. The membranes were blocked in 5% milk-TBST (50 mM Tris-Cl, pH 7.6; 150 mM NaCl, 0.1% tween-20) for 30 min. The primary antibody was added at 1:1000 dilution (unless otherwise stated) to 5% milk-TBST and rocked overnight at 4℃. The membrane was washed three times for 10 min each in TBST. Secondary antibody was added at 1:20,000 to TBST and rocked for 1 hr at room temperature. The membrane was washed three more times for 15 min each before chemiluminescent substrate was added and visualized using BioRad XRS imaging cabinet. Antibodies include a custom Mif1 antibody (Genscript), S-Tag antibody (GenScript catalog no. A00625, RRID:AB_915085), DYKDDDDK antibody (Thermo Fisher Scientific catalog no. 701629, RRID:AB_2532497), GFP antibody (Thermo Fisher Scientific catalog no. G10362, RRID:AB_2536526) and Goat anti-Rabbit IgG (H+L) secondary antibody, HRP (Thermo Fisher Scientific Catalog no. 31460, RRID:AB_228341).

## Electroporation

The method for electroporation of *Hydroides* larvae was adapted from those established for ascidian embryos (*Zeller, 2018*; *Zeller et al., 2006*). Specifically, 50 µl of 0.77 M mannitol, 20 µl of concentrated larvae (approximately 30 larvae), and 10 µl of purified protein (1.25–12.5 µg, 15.6–156 ng/µl final concentration based on protein recovery from inclusion bodies) were mixed and added to a 2 mm electroporation cuvette. The mixture was then electroporated with 30 V (150 V/cm) at 10 ohms and 3000 µF using a custom electroporation apparatus as previously described (*Zeller et al., 2006*). After electroporation, the mixture was immediately removed from the cuvette and mixed with 1 ml filtered artificial sea water and transferred into a 24-well plate. The larvae were then observed for metamorphosis 24 to 72 hr later (dependent on WT MACs positive control). Purification of proteins was performed on three separate occasions and each purification was electroporated twice, for a total of six independent biological replicates, each yielding similar outcomes.

## Quantification of electroporation

To test whether purified protein is transferred to tubeworm larvae by electroporation, larvae were concentrated to 30 larvae/µl. Final GFP concentration used was 0.625 µg/µl (50 µg total per electroporation). 20 µl of larvae were either electroporated by adding them to 50 µl 0.77 M mannitol and 10 µl of 5 µg/µl purified GFP, then electroporated at 30 V (150 V/cm) at 10 ohms and 3000 µF or not electroporated. Larvae were recovered from the electroporation cuvette by adding 1 ml instant ocean and moved to microcentrifuge tubes. Larvae were then washed 5 times with 1 ml instant ocean by first spinning down at 4000 g for 30 s and removing all but 50 µl of liquid. After the five washes larvae were then spun down at 4000 for 2 min and the sea water was removed. On ice, the larvae were then lysed in 100 µl 50 mM tris pH 8, 150 mM NaCl, 1% Triton X-100, and vortexed three times for 30 s each. Cell debris was pelleted by centrifuging at 21,000 g for 10 min and the pellet was discarded. The lysate was then quantified by first diluting a small aliquot of lysate (5 µl) 1:10 and using a Bradford protein assay. The GFP standard curve was created by performing 2-fold serial dilutions of the original purified GFP and using densitometry of the western blot. 8.76 µg of larvae lysate was loaded onto the same gel, with both the mock zap negative control and the electroporated larvae. Recovered protein was calculated using densitometry and the GFP standard curve. This was repeated for four biological replicates each with separate western blots.

## Pulldown assays

*E. coli* containing a dual expression plasmid with 12605, Mif1, or both 12605 and Mif1 was grown in 50 ml of LB supplemented with chloramphenicol (100 μg/ml) until an $OD_{600}$ 0.6 at 37℃. The *E. coli* plasmid was induced with 1 mM IPTG and the temperature was lowered to 20℃, then expression was allowed to proceed overnight (16 hr). Cells were recovered by pelleting at 4000 g for 10 min and media was discarded. Cells were resuspended in 25 mM tris pH 7.6, 150 mM NaCl and protease inhibitor cocktail (100 μM leupeptin, 1 μM pepstatin and 5 μM bestatin). Cells were lysed twice by French press 1000 Psi and then centrifuged at 10,000 g to remove cellular debris where the supernatant was then transferred to a new tube and the pellet was discarded. The conjugated agarose beads (Ni-NTA agarose and S-Tag binding agarose, EMD Millipore 69704–3) were equilibrated with the lysis buffer and 1 ml of agarose was added in batch to the recovered supernatant. Agarose was then recovered on a column and washed an additional two times with the lysis buffer to remove non-specific bound proteins. Proteins were eluted using either, 500 μl 3 M MgCl2, 20 mM tris pH 7.6 (S-TAG beads), or 500 μl 250 mM imidazole, 500 mM NaCl, 20 mM Tris pH 7.6 (Ni-NTA beads). 20 μg protein was loaded onto SDS-PAGE gel and then transferred onto PVDF membrane for western blot using either the endogenous Mif1 antibody or S-Tag antibody (GenScript Cat# A00625, RRID:AB_915085).

## Mass spectrometry

*P. luteoviolacea* was grown in 50 ml Marine Broth (MB) media in 250 ml flasks at 30℃ for 6 hr or overnight (12–14 hr). Cells were centrifuged for 30 min at 7000 g and 4℃ and resuspended in 5 ml cold extraction buffer (20 mM Tris, pH 7.5, 1 M NaCl). The resuspensions were centrifuged for 30 min at 4000 g and 4℃ and the supernatant was isolated and centrifuged for 30 min at 7000 g and 4℃. The pellet was resuspended in 20–100 μl cold extraction buffer and stored at 4℃ for further use. All mass spectrometry was done by the Functional Genomics Center Zurich (FGCZ). To prep the MAC extracts for mass spectrometry, the extracts were precipitated by mixing 30 μl of sample with 70 μl water and 100 μl 20% TCA. The samples were then washed twice with cold acetone. The dry pellets were dissolved in 45 μl buffer (10 mM Tris/2 mM CaCl$_2$, pH 8.2) and 5 μl trypsin (100 ng/μl in 10 mM HCl). They were then microwaved for 30 min at 60℃. The samples were dried, then dissolved in 20 μl 0.1% formic acid and transferred to autosampler vials for LC/MS/MS. 1 μl was injected.

## Acknowledgements

We thank Dr. Anca Segall and Dr. Manal Swairjo for reagents, technical support and constructive suggestions. We thank Ms. Amanda Alker and Ms. Nathalie Delherbe for their constructive suggestions. ScopeM at ETHZ and Ohad Medalia at the University of Zürich are acknowledged for instrument access. We thank Paula Picotti and Marco Faini for discussions of mass spectrometry experiments. This work was supported by the Harold and June Memorial Scholarship (CFE), Norma Sullivan Memorial Endowed Scholarship (CFE), Howard Hughes Medical Institute (DKN), NIH NIDCD (1R21DC013180-01A1, RWZ), Office of Naval Research (N00014-17-1-2677, NJS and SB), Office of Naval Research (N00014-16-1-2135, NJS), Office of Naval Research (N00014-14-1-0340, NJS and DKN), Alfred P Sloan Foundation, Sloan Research Fellowship (NJS), San Diego State University (NJS), European Research Council (679209, MP), Swiss National Science Foundation (31003A_179255, MP) and Gebert Rüf Foundation (MP).

## Additional information

### Competing interests

Charles F Ericson, Martin Pilhofer, Nicholas J Shikuma: has two provisional patents pending related to MACs in the 507 United States, Application Number: 62/768,240 and 62/844,988. The other authors declare that no competing interests exist.

## Funding

| Funder | Grant reference number | Author |
|---|---|---|
| Harold and June Memorial Scholarship | | Charles Ericson |
| Norma Sullivan Memorial Endowed Scholarship | | Charles Ericson |
| Howard Hughes Medical Institute | | Dianne K Newman |
| National Institute on Deafness and Other Communication Disorders | 1R21DC013180-01A1 | Robert W Zeller |
| Office of Naval Research | N00014-17-1-2677 | Nicholas J Shikuma |
| Office of Naval Research | N00014-16-1-2135 | Nicholas J Shikuma |
| Office of Naval Research | N00014-14-1-0340 | Dianne K Newman Nicholas J Shikuma |
| Alfred P. Sloan Foundation | | Nicholas J Shikuma |
| European Research Council | 679209 | Martin Pilhofer |
| Swiss National Science Foundation | 31003A_179255 | Martin Pilhofer |
| Gebert Rüf Stiftung | | Martin Pilhofer |

The funders had no role in study design, data collection and interpretation, or the decision to submit the work for publication.

## Author contributions

Charles F Ericson, Kyle E Malter, Formal analysis, Investigation, Visualization, Methodology, Writing—original draft, Writing—review and editing; Fabian Eisenstein, João M Medeiros, Formal analysis, Investigation, Visualization, Methodology, Writing—review and editing; Giselle S Cavalcanti, Investigation, Writing—review and editing; Robert W Zeller, Resources, Funding acquisition, Methodology, Writing—review and editing; Dianne K Newman, Conceptualization, Resources, Supervision, Funding acquisition, Validation, Project administration, Writing—review and editing; Martin Pilhofer, Conceptualization, Resources, Data curation, Formal analysis, Supervision, Funding acquisition, Investigation, Visualization, Methodology, Writing—original draft, Project administration, Writing—review and editing; Nicholas J Shikuma, Conceptualization, Resources, Supervision, Funding acquisition, Validation, Investigation, Visualization, Methodology, Writing—original draft, Project administration, Writing—review and editing

## Author ORCIDs

Charles F Ericson (iD) https://orcid.org/0000-0002-2854-0696
João M Medeiros (iD) https://orcid.org/0000-0001-9075-548X
Kyle E Malter (iD) https://orcid.org/0000-0002-3056-8751
Dianne K Newman (iD) http://orcid.org/0000-0003-1647-1918
Nicholas J Shikuma (iD) https://orcid.org/0000-0001-5518-5020

## Decision letter and Author response

Decision letter https://doi.org/10.7554/eLife.46845.023
Author response https://doi.org/10.7554/eLife.46845.024

# Additional files

## Supplementary files

• Transparent reporting form
DOI: https://doi.org/10.7554/eLife.46845.017

## Data availability

Subtomogram averages were deposited in the Electron Microscopy Data Bank (accession numbers EMD-4730 and EMD-4731).

The following datasets were generated:

| Author(s) | Year | Dataset title | Dataset URL | Database and Identifier |
|---|---|---|---|---|
| Fabian Eisenstein, Joao Medeiros, Martin Pilhofer | 2019 | Subtomogram average of MAC sheath and inner tube of P. luteoviolacea Mif1 mutant | https://www.ebi.ac.uk/pdbe/entry/emdb/EMD-4730 | Electron Microscopy Data Bank, EMD-4730 |
| Fabian Eisenstein, Joao Medeiros, Martin Pilhofer | 2019 | Subtomogram average of MAC sheath and inner tube of wildtype P. luteoviolacea | https://www.ebi.ac.uk/pdbe/entry/emdb/EMD-4731 | Electron Microscopy Data Bank, EMD-4731 |

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
