## [Decision Letter]

Thank you for submitting your article "A contractile injection system stimulates tubeworm metamorphosis by translocating a proteinaceous effector" for consideration by *eLife*. Your article has been reviewed by three peer reviewers, and the evaluation has been overseen by a Reviewing Editor and Gisela Storz as the Senior Editor. The reviewers have opted to remain anonymous.

The reviewers have discussed the reviews with one another and the Reviewing Editor has drafted this decision to help you prepare a revised submission.

Summary:

The manuscript by Ericson and colleagues investigates the Metamorphosis Associated Contractile (MAC) system that *Pseudoalteromonas luteoviolacea* employs to induce metamorphosis of host larvae of the tubeworm *Hydroides elegans*. The authors characterized a cluster of six genes, one of which, Mif1, was found to be essential for the functionality of the MAC system towards *Hydroides*. Structural analysis by electron-cryotomography and subtomogram averaging revealed that while in the wild-type MAC system there is a density inside the inner tube, the Mif1-deficient system is lacking this density, suggesting the location of the effector to be inside the inner tube. In agreement of Mif1 being the translocated effector, electroporation of Mif1 into tubeworm larvae was sufficient to trigger metamorphosis.

The manuscript is interesting and well written; however, several essential points have to be addressed.

Essential comments:

1) The reviewers think that the assignment of the localization of Mif1 in the MAC particle requires more validation. Biochemical techniques such as SDS-PAGE or Western blotting could be used to analyze the protein composition in purified WT and ΔJF50_12615 particles normalized to the same protein concentration. Furthermore, there is no interaction observed between Mif1 and the tube component by BTH. This could not be taken as conclusive, particularly since the gp19-like structure in the assembled MAC system is a hexamer in which Mif1 could be fitted similarly to the Tse2-Hcp1 interaction in T6SS (Silverman et al., 2013). The reviewers suggest that similar approaches be considered to test the fit of Mif1 within the MAC tube.

2) A standard test for validating the function of the protein is the complementation assay. While the functional complementation for JF50_12605 and JF50_12615 mutants is shown in Figure 1B, the reviewers wonder whether the tubes were also found full after complementation of JF50_12615?

3) Could the authors suggest a potential function or discuss functional units of JF50_12615 based on its the amino sequence or its location in the genome? Any conserved/hypervariable regions? Do *Photorhabdus* or *Serratia* contain orthologs? The authors suggest that Mif1 can make pores. Are there any hydrophobic domains that could be identified which would support this hypothesis? JF50_12615 occupies the central channel of the tube and thus it could be related to tape measure proteins. Do the WT and ΔJF50_12615 mutant have similar or different particle length distributions?

4) The authors show that a mutant lacking JF50_12605 is impaired in inducing metamorphosis but instead and in contrast to the Mif1 mutant, when extracellular MACs are prepared from the JF50_12605 mutant, there are no significant defect observed. There is no clear explanation on why this could be. Also there is no suggestion about the role of JF50_12605. Would that protein somehow be involved in helping loading Mif1 into the MACs, a kind of chaperone-like element? Is there any homology, or other features that stand out when examining the sequence of JF50_12605? The reviewers believe that this deserves a little bit more discussion, particularly because an interaction between Mif1 and JF50_12605 has been found using BTH. From this prospect, other methods should be used to validate this interaction, for example co-purification and pull down. This is even more true since the proportion of empty MACs appeared to be similar for the Mif1 and the JF50_12605 mutants.

5) The authors have debated whether the empty MACs could have been missing the inner tube homologous to the bacteriophage tube made of gp19 (JF50_12680). Would the MACs form in absence of the tube? If yes, how do MACs without a tube look like?

6) The authors report they have identified a second MACs effector, a nuclease they called Pne1 (preprint on bioRxiv). Are the authors suggesting that Pne1 is not in the lumen of the MACs? This is confusing and the relationship between Pne1 and Mif1 should be clarified either experimentally, i.e. by subtomogram averaging of Pne1-deficient MACs or in the Discussion.

7) To aid the interpretation of the electroporation experiment it would be great to know how much protein actually ends up in the worm (if any)? Can that amount be compared to the total protein content of the worm?

8) The quality of electron cryotomography and subtomogram averaging data is below the currently accepted standards in the field. The structures presented in Figure 2 have signs of overfitting as revealed by a texture outside of the protein density in the panel A. From the description in the Materials and methods section is not clear how the subtomogram alignment was performed, this should be briefly stated in the Materials and methods section additionally to the reference to Weiss et al. (2017). Each subtomogram average reconstruction has to have a corresponding resolution estimated by similarity between two statistically independently generated maps (so called "gold standard" processing). The similarity curves have to be presented in a supplementary figure. The pixel size at which the data is recorded is not needed in the figures. Figure 1—figure supplements 2 and 3 would benefit from noise reduction in the corresponding tomograms by i.e. non-linear anisotropic diffusion.

---

## [Author Response]

Essential comments:1) The reviewers think that the assignment of the localization of Mif1 in the MAC particle requires more validation. Biochemical techniques such as SDS-PAGE or Western blotting could be used to analyze the protein composition in purified WT and ΔJF50_12615 particles normalized to the same protein concentration. Furthermore, there is no interaction observed between Mif1 and the tube component by BTH. This could not be taken as conclusive, particularly since the gp19-like structure in the assembled MAC system is a hexamer in which Mif1 could be fitted similarly to the Tse2-Hcp1 interaction in T6SS (Silverman et al., 2013). The reviewers suggest that similar approaches be considered to test the fit of Mif1 within the MAC tube.

To address the reviewers’ comment, we generated strains with Mif1 (JF50-12615) tagged in three locations with FLAG epitopes in the native Mif1 chromosomal locus and performed western dot blot on purified MACs from each strain using an anti-FLAG antibody. We specifically detected the FLAG-tagged Mif1 proteins in each strain where Mif1 was tagged with FLAG. Results are now provided in Figure 3B and in the second paragraph of the subsection “The density within the MAC tube lumen represents a cargo protein”. This supports our imaging and mass spectrometry experiments and suggests that Mif1 is indeed found in the MAC structure.

Two different observations could explain the lack of interactions between Mif1 and the tube component. (1) We observed a low-density region between the density of the cargo and the density of the tube (shown by the black arrows in Figure 2B). (2) When expelled tubes were imaged by cryotomography, the tubes were always empty shown in the now added Figure 2—figure supplement 2. Together, these findings suggest that interactions between cargo and tube are weak or non-existent, which might facilitate the rapid release of the cargo into the target upon contraction (see subsection “The density within the MAC tube lumen represents a cargo protein”, first paragraph and Discussion, third paragraph).

As suggested, we purified heterologously-expressed tube protein and analyzed it by negative stain electron microscopy. Under the tested conditions, however, no ring structures were observed, which prevented us from performing the subsequent experiments according to Silverman et al. (2013).

2) A standard test for validating the function of the protein is the complementation assay. While the functional complementation for JF50_12605 and JF50_12615 mutants is shown in Figure 1B, the reviewers wonder whether the tubes were also found full after complementation of JF50_12615?

We visualized MACs from the ΔJF50_12605::JF50_12605 and Δ*mif1::mif1* strains and found that the inner tubes of both complemented strains showed a filled phenotype. These results are now included in the text (subsection “Two bacterial genes are responsible for densities within the inner tube lumen of MACs and are involved in metamorphosis induction”, last paragraph) and Figure 1—figure supplement 4.

3) Could the authors suggest a potential function or discuss functional units of JF50_12615 based on its the amino sequence or its location in the genome? Any conserved/hypervariable regions? Do Photorhabdus or Serratia contain orthologs? The authors suggest that Mif1 can make pores. Are there any hydrophobic domains that could be identified which would support this hypothesis? JF50_12615 occupies the central channel of the tube and thus it could be related to tape measure proteins. Do the WT and ΔJF50_12615 mutant have similar or different particle length distributions?

We thoroughly searched for domains that would yield clues to Mif1’s (JF50_12615’s) mechanism of inducing metamorphosis. Unfortunately, we did not identify any functional or transmembrane domains in Mif1. Other bacteria like *Photorhabdus* or *Serratia* do not contain orthologs of Mif1. This is consistent with the fact that contractile injection systems from these organisms are not known to induce metamorphosis. The Mif1-genomic context does not suggest a mechanism of function either. We have modified the Discussion to make our conclusions more clear (Discussion, second paragraph).

Mif1 does not show sequence similarities to tape measure proteins. Wildtype and ∆*mif1* MACs did also not show differences in length distribution (see also Figure 1—figure supplement 2 and 3)

4) The authors show that a mutant lacking JF50_12605 is impaired in inducing metamorphosis but instead and in contrast to the Mif1 mutant, when extracellular MACs are prepared from the JF50_12605 mutant, there are no significant defect observed. There is no clear explanation on why this could be. Also there is no suggestion about the role of JF50_12605. Would that protein somehow be involved in helping loading Mif1 into the MACs, a kind of chaperone-like element? Is there any homology, or other features that stand out when examining the sequence of JF50_12605? The reviewers believe that this deserves a little bit more discussion, particularly because an interaction between Mif1 and JF50_12605 has been found using BTH. From this prospect, other methods should be used to validate this interaction, for example co-purification and pull down. This is even more true since the proportion of empty MACs appeared to be similar for the Mif1 and the JF50_12605 mutants.

We agree that the role of JF50_12605 deserves more attention. In regards to metamorphosis induction by ∆JF50_12605 strains, the discrepancy between bacterial biofilms and purified MACs is likely based on higher concentrations of MACs affecting the larvae when MAC extracts are used. In mass spectrometry analysis of ∆JF50_12605 MACs a low abundance of Mif1 was observed (Figure 3A) suggesting that Mif1 may be able to associate in rare cases with MACs independent of JF50_12605. We have revised the text (subsection “The density within the MAC tube lumen represents a cargo protein”, Discussion, third paragraph) to explain this.

In order to validate the interaction between Mif1 and JF50_12605, we performed the suggested reciprocal pull-down of JF50_12605-S-tag with Mif1-6xHis-tag. We detected JF50_12605 by western blot after pull-down (now Figure 3C, subsection “The density within the MAC tube lumen represents a cargo protein”, last paragraph), corroborating our bacterial two-hybrid analyses (Figure 3C-F).

5) The authors have debated whether the empty MACs could have been missing the inner tube homologous to the bacteriophage tube made of gp19 (JF50_12680). Would the MACs form in absence of the tube? If yes, how do MACs without a tube look like?

We showed in a previous paper that a mutant lacking the tube protein does not assemble MACs (Shikuma et al., 2014). In the previous version of the presented manuscript, we docked a homology model of the gp19 tube protein into the EM densities, in order to unambiguously show that the cargo density within the tube lumen was distinct from the tube itself. We realize that the way this was discussed might have been confusing and revised the text and figure (subsection “The density within the MAC tube lumen represents a cargo protein”, first paragraph, Figure 2).

6) The authors report they have identified a second MACs effector, a nuclease they called Pne1 (preprint on bioRxiv). Are the authors suggesting that Pne1 is not in the lumen of the MACs? This is confusing and the relationship between Pne1 and Mif1 should be clarified either experimentally, i.e. by subtomogram averaging of Pne1-deficient MACs or in the Discussion.

We agree that the localization of the second effector, Pne1, requires additional discussion. We have included the cryotomogram of the ∆*pne1* (JF50_12610) mutant in Figure 1—figure supplement 3 and show that the tube has a filled phenotype. The filled phenotype suggests that Pne1 is not found within the inner tube, but instead could be loaded in a different location (e.g. spike) within the MACs complex. We have modified the labeling of Figure 1—figure supplement 3 to indicate that JF50_12610 is named Pne1 and added text to discuss this point (Discussion, fourth paragraph). The Pne1 paper has been recently published (Rocchi et al., 2019).

7) To aid the interpretation of the electroporation experiment it would be great to know how much protein actually ends up in the worm (if any)? Can that amount be compared to the total protein content of the worm?

To determine if electroporation facilitates protein association with tubeworm larvae, we electroporated larvae in the presence of GFP protein, thoroughly washed the larvae and performed western blot analysis. We found that larvae subjected to electroporation had a higher concentration of GFP associated when compared to non-electroporated larvae. These results are now included in the manuscript (subsection “Purified and electroporated Mif1 protein induces tubeworm metamorphosis” and as Figure 4—figure supplement 2).

8) The quality of electron cryotomography and subtomogram averaging data is below the currently accepted standards in the field. The structures presented in Figure 2 have signs of overfitting as revealed by a texture outside of the protein density in the panel A. From the description in the Materials and methods section is not clear how the subtomogram alignment was performed, this should be briefly stated in the Materials and methods section additionally to the reference to Weiss et al. (2017). Each subtomogram average reconstruction has to have a corresponding resolution estimated by similarity between two statistically independently generated maps (so called "gold standard" processing). The similarity curves have to be presented in a supplementary figure. The pixel size at which the data is recorded is not needed in the figures. Figure 1—figure supplements 2 and 3 would benefit from noise reduction in the corresponding tomograms by i.e. non-linear anisotropic diffusion.

Subtomogram averages were recalculated using the same raw data and initial particle coordinates. Instead of PEET, Dynamo was used and data sets were split according to the “gold standard” before calculating FSC curves (now shown in Figure 2—figure supplement 1). Materials and methods were updated to describe the averaging procedure in detail.

The subtomogram average of ∆JF50_12585 was removed, since it did not add any additional insight to this study.

Pixel sizes were removed from all figure legends.

Figure 1—figure supplement 2 was filtered using a deconvolution filter.

Figure 1—figure supplement 3 was adapted to show density plots that clearly indicate filled and empty phenotypes.